# Long-Term Experience with Hyperthermic Chemotherapy (HIVEC) Using Mitomycin-C in Patients with Non-Muscle Invasive Bladder Cancer in Spain

**DOI:** 10.3390/jcm10215105

**Published:** 2021-10-30

**Authors:** Ana Plata, Félix Guerrero-Ramos, Carlos Garcia, Alejandro González-Díaz, Ignacio Gonzalez-Valcárcel, José Manuel de la Morena, Francisco Javier Díaz-Goizueta, Julio Fernández del Álamo, Victoria Gonzalo, Javier Montero, Alejandro Sousa-Escandón, Juan León, Jose Luis Pontones, Francisco Delgado, Miguel Adriazola, Ángela Pascual, Jesús Calleja, Ana Ruano, Luis Martínez-Piñeiro, Javier C. Angulo

**Affiliations:** 1Urology Department, Hospital Universitario de Canarias, Carretera Ofra s/n, 38320 San Cristóbal de La Laguna, Spain; anaplatabello@hotmail.com (A.P.); carlosgcruza@gmail.com (C.G.); 2Urology Department, Hospital Universitario 12 de Octubre, Avenida de Córdoba s/n, 28041 Madrid, Spain; felixguerrero@gmail.com (F.G.-R.); alejandroglezdiaz@gmail.com (A.G.-D.); 3Urology Department, Hospital Universitario Infanta Sofía, Paseo de Europa 34, San Sebastián de los Reyes, 28702 Madrid, Spain; ignacio.gonzalezvalcarcel@salud.madrid.org (I.G.-V.); josemanuel.morena@salud.madrid.org (J.M.d.l.M.); 4Urology Department, Hospital Universitario Getafe, Carretera de Toledo, Km 12.500, Getafe, 28905 Madrid, Spain; jdgoizueta@hotmail.com; 5Urology Department, Hospital Universitario de Torrejón, Mateo Inurria, s/n, Torrejón de Ardoz, 28850 Madrid, Spain; jfdel@torrejonsalud.com; 6Urology Department, Hospital Universitario de Burgos, Avenida Islas Baleares 3, 09006 Burgos, Spain; vgolzalo@saludcastillayleon.es (V.G.); jmonterot@saludcastillayleon.es (J.M.); 7Urology Department, Hospital Comarcal de Monforte, Rúa Corredoira s/n, 27400 Monforte de Lemos, Spain; sousa-alejandro@hotmail.com (A.S.-E.); juanleonmata@gmail.com (J.L.); 8Urology Department, Hospital Universitario La Fe, Avinguda de Fernando Abril Martorell 106, 46026 Valencia, Spain; pontones_jos@gva.es (J.L.P.); delgado_fra@gva.es (F.D.); 9Urology Department, Hospital General Rio Carrión, Avenida Donantes de Sangre s/n, 34005 Palencia, Spain; madriazola@telefonica.net (M.A.); pascual-uro@hotmail.com (Á.P.); 10Urology Department, Hospital Clínico Universitario de Valladolid, Av. Ramón y Cajal 3, 47003 Valladolid, Spain; jcalleja@saludcastillayleon.es (J.C.); aruanom@saludcastillayleon.es (A.R.); 11Urology Department, Hospital Universitario La Paz, Paseo de la Castellana 261, 28046 Madrid, Spain; luis.mpineiro@salud.madrid.org; 12Clinical Department, Facultad de Ciencias Biomédicas, Universidad Europea de Madrid, Carretera de Toledo, Km 12.500, Getafe, 28905 Madrid, Spain

**Keywords:** bladder neoplasia, hyperthermic intravesical chemotherapy, mitomycin-C, bladder recirculation system

## Abstract

(1) Background: Intravesical mitomycin-C (MMC) combined with hyperthermia is increasingly used in non-muscle invasive bladder cancer (NMIBC), especially in the context of a relative BCG shortage. We aim to determine real-world data on the long-term treatment outcomes of adjunct hyperthermic intravesical chemotherapy (HIVEC) with MMC and a COMBAT® bladder recirculation system (BRS); (2) Methods: A prospective observational trial was performed on patients with NMIBC treated with HIVEC using BRS in nine academic institutions in Spain between 2012–2020 (HIVEC-E). Treatment effectiveness (recurrence, progression and overall mortality) was evaluated in patients treated with HIVEC MMC 40mg in the adjuvant setting, with baseline data and a clinical follow-up, that comprise the Full Analysis Set (FAS). Safety, according to the number and severity of adverse effects (AEs), was evaluated in the safety (SAF) population, composed by patients with at least one adjunct HIVEC MMC instillation; (3) Results: The FAS population (*n* = 502) received a median number of 8.78 ± 3.28 (range 1–20) HIVEC MMC instillations. The median follow-up duration was 24.5 ± 16.5 (range 1–81) months. Its distribution, based on EAU risk stratification, was 297 (59.2%) for intermediate and 205 (40.8%) for high-risk. The figures for five-year recurrence-free and progression-free survival were 50.37% (53.3% for intermediate and 47.14% for high-risk) and 89.83% (94.02% for intermediate and 84.23% for high-risk), respectively. A multivariate analysis identified recurrent tumors (HR 1.83), the duration of adjuvant HIVEC therapy <4 months (HR 1.72) and that high-risk group (HR 1.47) were at an increased risk of recurrence. Independent factors of progression were high-risk (HR 3.89), recurrent tumors (HR 3.32) and the induction of HIVEC therapy without maintenance (HR 2.37). The overall survival was determined by patient age at diagnosis (HR 3.36) and the treatment duration (HR 1.82). The SAF population (*n* = 592) revealed 406 (68.58%) patients without AEs and 186 (31.42%) with at least one AE: 170 (28.72%) of grade 1–2 and 16 (2.7%) of grade 3–4. The most frequent AEs were dysuria (10%), pain (7.1%), urgency (5.7%), skin rash (4.9%), spasms (3.7%) and hematuria (3.6%); (4) Conclusions: HIVEC using BRS is efficacious and well tolerated. A longer treatment duration, its use in naïve patients and the intermediate-risk disease are independent determinants of success. Furthermore, a monthly maintenance of adjunct MMC HIVEC diminishes the progression rate of NMIBC.

## 1. Introduction

Bladder cancer is a major urological disease, with more than half a million new cases diagnosed and leading to almost 0.2 million deaths per year worldwide [1]. An increase in bladder cancer incidence but a decrease in mortality has been recently observed in several European countries, possibly related to a better awareness and earlier detection that allows better oncological control [2]. In Spain, despite the efforts launched to limit smoking habit, age-standardized incidence and mortality rates remain at 15.6 and 3.5 per 100.000, respectively [3]. Approximately 75% of patients present as having non-muscle invasive bladder cancer (NMIBC), which, despite being a none life-threatening disease, presents the risk of recurrence and also of progression to a muscle invasive form, most often leading to metastases [4].

Treatments to limit the recurrence and progression of NMIBC include intravesical mitomycin C (MMC) and bacillus Calmette-Guérin (BCG). Maintenance BCG is considered to be the best bladder-sparing treatment for high-risk NMIBC patients and also as a potential alternative for the intermediate risk group [5]. However, BCG administration is by far more toxic than chemotherapy [6,7]. Additionally, recent evidence suggests that maintenance BCG is not a cost-effective alternative for the entire population of patients with intermediate/high risk NMIBC [8]. To make matters worse, shortages in the BCG supply have compromised patient outcomes and left clinicians around the globe without clear effective and reliable alternatives [9,10,11].

Despite conducting instillations with MMC and other chemotherapeutic agents for decades, the length and frequency of the adjuvant chemotherapy regime is controversial and has yet to be established [5]. The Global BCG shortage has not been completely resolved, leading to the need to urgently develop strategies to improve the efficacy of chemotherapy delivery. The concept of device-assisted intravesical therapy to improve the penetration of MMC and other chemotherapeutic agents into the bladder wall is very promising. Two different heating systems, microwave-induced chemo-hyperthermia using radiofrequency (RF) and hyperthermic intravesical chemotherapy (HIVEC) using the bladder recirculation system (BRS), have been increasingly used for both intermediate and high-risk patients during BCG shortage, sometimes without a solid grounding due to the absence of clinical trials [12]. Neither the optimal regime for standard chemotherapy instillations nor the optimal regime for device-assisted chemotherapy have been identified. 

We present real-world oncological results of chemo-hyperthermia using Combat BRS from a prospective observational trial conducted in Spain (HIVEC-E). The main objective of the study is to assess the safety and effectiveness results of the therapy in a real world setting and assist in identifying the optimal regime that should be considered when designing new randomized clinical trials using HIVEC. 

## 2. Materials and Methods

### 2.1. Study Population

A prospective observational multicentre study was performed on consecutive patients with NMIBC, treated with HIVEC using the COMBined Antineoplastic Thermotherapy (COMBAT®) BRS (Combat Medical, Wheathampstead, UK) in nine academic institutions in Spain between 2012 and 2020. The registry, named the HIVEC-epidemiology (HIVEC-E), included consecutive patients with NMIBC, treated with any form of chemotherapeutic regime using the COMBAT BRS device, that increases the temperature of the pharmaceutical agent to 43 °C (± 1 °C) outside the body to enter the bladder through a soft 16F 3-way Foley catheter and recirculate with a constant flow for a period of 1 hour. A closed circuit with a heating system keeps the chemotherapy at a constant temperature. At the end of the procedure, the product is collected in a urine collection bag.

All patients provided their informed consent to participate in the study. Investigators registered the clinical information into an electronic case report form (eCRF) with periodical status updates during follow-up. All patients were treated with a complete transurethral resection of the bladder (TURB) and HIVEC using different agents in the adjuvant (prophylactic) or neoadjuvant (ablative) setting, with or without maintenance, according to the decision made by the investigators and the regular practice in their institution. Histopathologic evidence of muscle-invasive disease was excluded in every case. Patients not included in this trial were treated according to European Guidelines whenever treatment was available.

The Full Analysis Set (FAS) population for the current study focused exclusively on patients receiving adjunct HIVEC MMC 40mg (standard dose) and a clinical follow-up was updated in June 2021. The treatment modalities included in this study were based on weekly induction HIVEC MMC, either as a one-time treatment or followed by monthly maintenance, or as monthly maintenance alone for other patients. Patients receiving a combination treatment with induction BCG and monthly HIVEC MMC maintenance because of BCG shortage during the treatment were excluded. The number of instillations varied according to the criteria of the different centers involved, the patient risk group and treatment tolerance. The Safety (SAF) population was defined as all subjects who received at least one adjunct HIVEC MMC instillation with a post-baseline safety assessment. The statement that a subject had no adverse events (AEs) also constitutes a safety assessment. The number and severity of AEs were evaluated and were, according to Common Terminology Criteria for Adverse Events (CTCAE), defined as Grade 1 (mild toxicity), Grade 2 (moderate), Grade 3 (severe), Grade 4 (life-threatening) and Grade 5 (death). For practical purposes, the mild-moderate and severe-life threatening toxicities were pooled together.

### 2.2. Study Endpoints

The co-primary endpoints were the evaluation of the effectiveness of adjuvant HIVEC MMC in the FAS population in terms of recurrence-free, progression-free and overall survival. A multivariate analysis was performed to determine prognostic factors, and thus evaluate the likely markers of treatment response. The secondary endpoint was the evaluation of safety of the adjuvant HIVEC MMC in the SAF population. 

### 2.3. Variables Evaluated

Data registered in eCRF included patients’ baseline patient characteristics (date of birth, sex, body mass index (BMI), smoking habit), former tumor history (previous tumors, treatments received), date of inclusion, preoperative tumor characteristics (tumour size, multiplicity), operative data (date of TURB, 2nd TURB, bladder biopsy mapping) and postoperative data (T category, tumor grade, presence of concomitant cis, EAU risk group), treatment schedule (date of HIVEC instillation, tolerance), status at each follow-up visit (date of event, recurrence, progression, cystectomy) and patient death. Cause of death was registered whenever possible but was preferred overall to disease-specific survival as an endpoint in the absence of a mortality committee.

### 2.4. Statistical Analysis

Mean and standard deviation (SD) or median and interquartile range (IQR) were calculated for quantitative variables and those that were qualitative were described using absolute and relative frequencies. A paired t-test or Wilcoxon rank sum test were used to compare continuous variables. A Cochran–Armitage trend test and Chi-square contingency test or a Fisher exact test were performed to compare the categorical variables. The factors affecting tumor recurrence, progression and overall survival were evaluated using the Kaplan-Meier analysis method and their significance was evaluated by two-sided log-rank test. All patients were updated to June 2021. A univariate analysis using hazard ratios and 95% Wald confidence limits was performed for the variables investigated. All the variables with a significant impact in the univariate analysis were evaluated in a multivariate Cox regression model using a stepwise logistic regression with *p* = 0.15 entry and *p* = 0.05 stay criteria. Both the hazard ratio and 95% confidence intervals were calculated for the multivariate models defining disease recurrence, progression and overall survival. A *p*-value of <0.05 was considered as significant. The statistical analysis was performed using Statistical Analysis System 9.4 (SAS Institute Inc, Cary, NY, USA).

## 3. Results

Figure 1 shows the flowchart of patients registered in HIVEC-E and the populations of patients included in this analysis. The FAS population (*n* = 502) included patients receiving adjunct HIVEC MMC with follow-up treatment that allowed for the evaluation of primary endpoints (recurrence-free, progression-free and overall survival).

Patient and tumor characteristics are summarized in Table 1. The distribution, based on EAU risk stratification, was 297 (59.2%) for intermediate and 205 (40.8%) for high-risk. Mapping bladder biopsies were performed at the time of TURB in 45 patients and revealed carcinoma in situ (cis) in 22 (4.4%); primary cis in 10 (2%) and cis concomitant to papillary neoplasia in 12 (4.4%).A second-TURB was performed in 122 (24.3%) and revealed persistent NMIBC in 11 (2.2%). According to the previous tumor history and treatments received, a tumor was recurrent in 214 patients (42.6%) and in 62 (12.35%) patients the recurrence rate was higher than 1 episode per year. Regarding previous treatments received, none of the patients had been treated with device-assisted intravesical therapies before their inclusion in HIVEC-E, however, 69 (13.7%) had received MMC in normothermia and 52 (10.4%) received BCG before their inclusion.

A median number of 8.91 ± 3.22 (range 1–20) HIVEC MMC instillations per patient were administered in the FAS population; 4.63 ± 1.68 for patients treated with a weekly induction schedule (*n* = 68), 8.7 ± 3.15 instillations for those treated with monthly maintenance alone (*n* = 27), and 9.64 ± 2.85 instillations for patients receiving both the weekly induction and monthly maintenance (*n* = 407). Globally, 434 (86.45%) patients received some form of a maintenance regime; however, treatment duration lasted for more than 4 months for only 371 patients (73.9%). The median follow-up was at 24.5 ± 16.5 (range 1–81) months. During follow-up, 159 patients (31,7%) suffered disease recurrence, 35 (7%) patients progressed to a muscle invasive disease and 66 (13.5%) died (any cause). A radical cystectomy was performed as a rescue surgery in 22 of the 35 patients with disease progression (62.9%). In no case did a cystectomy present as technically more challenging. Furthermore, a cystectomy was performed in another case without neoplasia due to a retractile bladder after repeated TURB. 

Figure 2 shows the Kaplan-Meir curves for the recurrence-free interval, progression-free interval and overall survival for the FAS population, and also the stratification for the EAU intermediate- and high-risk groups evaluated.

Table 2 shows recurrence, progression and overall mortality at different times with interval limits for the FAS population and a stratification according to the risk groups, with a log-rank test for comparisons. The five-years recurrence-free survival rate was 50.37% for the total series (53.3% intermediate and 47.14% high-risk; log-rank, *p* = 0.075). Five-years progression-free survival was 89.83% (94.02% intermediate and 84.23% high-risk; log-rank, *p* = 0.001). The rate of five-years overall survival was 66.35% (74.26% intermediate and 60.12% high-risk; log-rank, *p* = 0.064). Among the high-risk group, the primary cis population (*n* = 10) revealed a 50% response rate and an 87.5% progression-free survival at 1 year; a 25% response rate and 65.6% progression-free survival at 2 years, etc. None of these patients died during follow-up due to intensive surveillance and rescue surgery.

### 3.1. Recurrence-Free Survival

Kaplan-Meier analysis revealed that T category (log-rank; *p* = 0.0004), presence of cis (log-rank; *p* = 0.0005), primary vs. recurrent tumor (log-rank; *p* = 0.0004), duration of treatment (log-rank; *p* = 0.0002), use of maintenance therapy (log-rank; *p* = 0.0007), previous treatment with MMC (log-rank; *p* = 0.0201) and previous treatment with BCG (log-rank; *p* = 0.0052) were predictors of tumor recurrence-free interval. Duration of HIVEC MMC (log-rank, *p* = 0.0002) seems more determinant than use of maintenance (log-rank, *p* = 0.0007) in terms of recurrence-free survival (Figure 3).

Table 3 shows the corresponding hazard ratios and confidence interval limits for each variable as obtained in the univariate analysis. The risk-group, T category, grade, cis, tumor history, duration of treatment, use of maintenance therapy, former use of MMC and of BCG were entered into the stepwise model for recurrence (*p* < 0.15). Patient age, sex, smoking habit, tumor multiplicity and tumor size were not related to tumor recurrence. A multivariate analysis revealed previous tumor history (recurrent vs. primary; HR 1.828 (95% CI 1.327–2.518); *p* = 0.0002), duration of treatment (<4 months vs. ≥4 months; HR 1.724 (95% CI 1.235–2.407); *p* = 0.0014) and EAU risk-group (high-risk vs. intermediate-risk; HR 1.472 (95% CI 1.071–2.024); *p* = 0.0171) remained independent factors (*p* < 0.05) of tumor recurrence using adjunct HIVEC MMC. 

### 3.2. Progression-Free Survival

Regarding progression to muscle invasive disease, a Kaplan-Meier analysis revealed that the EAU risk-group (log-rank; *p* = 0.001), T category (log-rank; *p* = 0.0004), presence of cis (log-rank; *p* = 0.0007), primary vs. recurrent tumor (log-rank; *p* = 0.0019), use of maintenance therapy (log-rank; *p* = 0.0016), previous treatment with MMC (log-rank; *p* = 0.0117) and previous treatment with BCG (log-rank; *p* = 0.0097) were predictive factors. The use of maintenance (log-rank; *p* = 0.0016) seems more determinant than the duration of the treatment (log-rank; *p* = 0.065) in terms of progression-free survival (Figure 3). Table 4 shows the univariate Cox regression analysis with hazard ratios for the variables evaluated. 

Patient sex, smoking habit, tumor multiplicity and tumor size did not appear related to tumor progression to the invasive disease. Conversely, patient age, EAU risk-group, T category, tumor grade, cis, tumor history, duration of treatment, use of maintenance therapy, former use of MMC and of BCG were entered into the stepwise model as likely determinant factors (*p* < 0.15). A multivariate analysis revealed that the EAU risk-group (high-risk vs. intermediate-risk; HR 3.891 (95% CI 1.886–8); *p* = 0.0002), previous tumor history (recurrent vs. primary; HR 3.32 (95% CI 1.613–6.833); *p* = 0.0011) and treatment schedule using maintenance (w/o vs. with maintenance; HR 2.374 (95% CI 1.125–5.01); *p* = 0.0233) independently predict progression to muscle invasive disease in patients receiving adjunct HIVEC with MMC in the present study. 

### 3.3. Overall Survival

A Kaplan-Meier analysis revealed that patient age (log-rank; *p* < 0.0001), tumor grade (log-rank; *p* = 0.046) and duration of treatment (log-rank; *p* = 0.0087) were predictive factors of mortality. Table 5 shows the univariate Cox regression analysis. The sex of the patient, smoking habit, presence of cis, previous history, tumor multiplicity, tumor size, previous treatment with MMC and with BCG are not related to mortality. However, patient age, EAU risk-group, T category, grade, concomitant cis, tumor history, duration of treatment, use of maintenance therapy, former use of MMC and of BCG were entered into the stepwise model as likely determinant factors (*p* < 0.15). A multivariate analysis revealed that patient age (older than 70 vs. 70 or less; HR 3.356 (95% CI 1.884–5.976); *p* < 0.0001) and treatment duration (<4 months vs. ≥4 months; HR 1.824 (95% CI 1.095–3.039); *p* = 0.0211) independently predict the survival (*p* < 0.05) of patients with NMIBC treated with adjunct HIVEC MMC.

### 3.4. Tolerability and Safety

The frequency and severity of AEs was assessed in the SAF population (*n* = 592), that included patients with at least one HIVEC MMC instillation on any schedule. Globally, 406 patients (68.58%) did not suffer any AEs while 186 (31.42%) registered at least one. A single AE was registered for 130 cases, 2 AES per patient for 36, 3 AES per patient for 16 and 4 AES per patient for 4. In total, the number of AES registered was 266 for 186 patients. The AE severity was evaluated as grade 1–2 in 170 patients (28.72%) and grade 3–4 in the remaining 16 (2.7%). No case revealed toxicity grade 5. Table 6 shows the distribution of severity of AEs according to each particular AE. The most frequent AEs were dysuria (9.9%), bladder pain (7.1%), urgency (5.7%), skin rash (4.9%), spasms (3.7%) and hematuria (3.55%). 

## 4. Discussion

Hyperthermia-based therapy for NMIBC is gaining traction, especially since the shortage of BCG has severely affected clinical practice in disease management worldwide. The problem of BCG shortage, and also of MMC more recently, appears to have been exacerbated in the COVID-19 pandemic, so finding a solution to the challenges in the optimalization of intravesical chemotherapy instillations is of supreme importance. In general, device-assisted therapies have gained popularity and, despite clinical evidence still not being mature, they constitute an attractive alternative to improve the efficacy of intravesical chemotherapy by enhancing cell membrane permeability to facilitate a higher penetration of the drug into the bladder and also for the direct toxic effect of heat [13,14,15]. Additionally, the release of the heat shock protein from cancer cells by chemo-hyperthermia could activate the adaptive T-cell response [16,17]. This presumed synergistic effect of hyperthermia and chemotherapy was demonstrated in vitro for several chemotherapeutic agents including MMC, epirubicin and gemcitabine [18]. 

The most common application of chemo-hyperthermia is as an adjuvant treatment (prophylactic) after complete TURB, with the intention to reduce the chance of tumor recurrence and progression. However, a neoadjuvant (ablative) approach can also be used in cases with a residual tumor after TURB and also for carcinoma in situ [19]. Different hyperthermia systems are available to heat the bladder, including microwave induced heating using an intravesical radiofrequency-emitting antenna incorporated in a catheter, conductive-based heating outside the bladder using a recirculating fluid system and an external radiofrequency energy source. The COMBAT ® BRS device uses a conductive aluminum heat exchanger that heats and controls the temperature at 43 °C. The first in vivo studies were conducted in the porcine model [20]. The preliminary clinical data, obtained using this system support, have shown satisfactory results both in the neoadjuvant and the adjunct setting [21,22]. Prospective trials have been specifically conducted in different populations that are currently under analysis. Highly interesting results have recently confirmed that HIVEC MMC is valuable in the high-risk NMIBC population [23,24] and also in cases of BCG failure [25,26]. The present study, based on a real-world analysis, provides additional prognostic information on the value of adjunct HIVEC MMC, both in intermediate and high-risk groups, and aims to provide a rationale for selecting specific populations that could benefit from this approach.

It is difficult to provide an indirect comparison of efficacy in the absence of control arms but the progression-free data we provide for HIVEC using BRS seem preferable to the results provided by the long-term experience with RF-induced hyperthermia combined with intravesical chemotherapy in the recent publication of Brummelhuis et al [27]. Similarly, the progression-free rate we report may also be equivalent to the results provided by long-term BCG maintenance [7,28]. However, a strict randomized comparison is necessary. A randomized study (HYMN trial) comparing RF-induced chemo-hyperthermia using MMC (6-weekly induction instillations, followed by maintenance instillations at 6-week intervals for the first year and at 8-week intervals for the second year) and BCG (induction and maintenance for one year) in patients with recurrent intermediate- and high-risk NMIBC following induction and/or maintenance BCG revealed no differences in complete response at both 3 months and in disease-free survival between the two groups [29]. The subgroup analyses in this trial have shown that patients with CIS had a lower disease-free survival with chemo-hyperthermia. Hopefully, trials with conductive chemo-hyperthermia will provide additional insight on the issue. In our experience, patients without cis, who are treated with adjunct HIVEC MMC, fare better than those with cis, both in terms of recurrence/tumor response and progression. However, in a multivariate analysis detection of cis, it is not found to be an independent predictor itself. 

The current study provides real world long-term data on COMBAT BRS adjunct HIVEC MMC. In an overt clinical practice setting, both a 50.37% recurrence-free (53.3% intermediate- and 47.14% high-risk) and 89.83% (94.02% intermediate- and 84.23% high-risk) progression-free survival rate at 5 years have been observed. This treatment is generally well tolerated, with 68.58% of patients suffering no AE, and a serious toxicity is presented in only 2.7%. No case presented life-threatening toxicity. Furthermore,, this treatment did not compromise the oncological outcomes of cystectomy in cases with disease progression after HIVEC. The safety data we confirm corresponds with other recently reported experiences from other countries [30,31]. 

The global tolerability of HIVEC-MMC seems to be much better than that of RF-induced hyperthermia in which a rate of 94.2% patients experience at least one AE [27]. Furthermore, the frequency of cutaneous contact allergic reactions observed with HIVEC MMC, including vesicular dermatitis of the hands and feet and/or dermatitis of the genitals, or even more widespread eruptions, is notably lower than the 15.4% rate recently reported for RF-induced hyperthermia [27].

The prospective nature of this study and the consecutive inclusion of patients reduces the risks of a selection bias, information bias and the underreporting of side effects, and brings uniformity both in the chemotherapy dosage and the technique of instillation with the BRS. However, the lack of a control group (MMC in normothermia or BCG) is a very serious limitation in our study. Additionally, the heterogeneous population included as the adjunct HIVEC MMC implies some variation in the maintenance scheme and treatment duration. Nevertheless, this has allowed us to discover the very interesting finding that monthly adjunct MMC HIVEC maintenance diminishes both the recurrence and progression rate of NMIBC compared to patients in which only a weekly induction regime has been applied. Use of a maintenance schedule is an independent factor to protect for its progression to a muscle invasive disease. However, the optimal regime for maintenance therapy, especially for the intermediate-risk group, is not well defined. 

Also, we confirm that for a longer treatment duration, both primary tumors and intermediate-risk disease have more favorable results both in recurrence and progression. It seems of paramount importance to define that adjunct COMBAT BRS treatment with MMC with a duration longer than 4 months is an independent prognostic factor, not only to prevent tumor recurrence but also in terms of mortality; and this finding appears critical for the design of comparative trials in the future. The long-term experience with RF-induced hyperthermia also confirms that a long-term maintenance confers better results [27].

The use of device-assisted intravesical therapy in patients in which former therapies, and more specifically BCG, have failed is an area of current and intensive investigation [32]. An additional finding in this COMBAT BRS adjunct MMC study is that the primary untreated patients benefit most from chemo-hyperthermia, a finding that is in consonance with the fact that nonprimary NMIBC also has poorer results for BCG [33]. Finally, patients at an intermediate-risk have a better response in terms of recurrence and progression than their high-risk counterparts; this finding should be analyzed and taken into account when clinical trials are designed for specific populations. 

## 5. Conclusions

HIVEC through MMC using COMBAT BRS is an efficacious and well-tolerated alternative for patients with intermediate- and high-risk NMIBC. A maintenance schedule should be recommended as the duration of treatment is the most important independent prognostic factor. Additionally, HIVEC MMC in primary tumors and intermediate-risk patients offers the best treatment results, and this could be taken into account when the use of long-term BCG requires optimization due to shortages. 

## Figures and Tables

**Figure 1 jcm-10-05105-f001:**
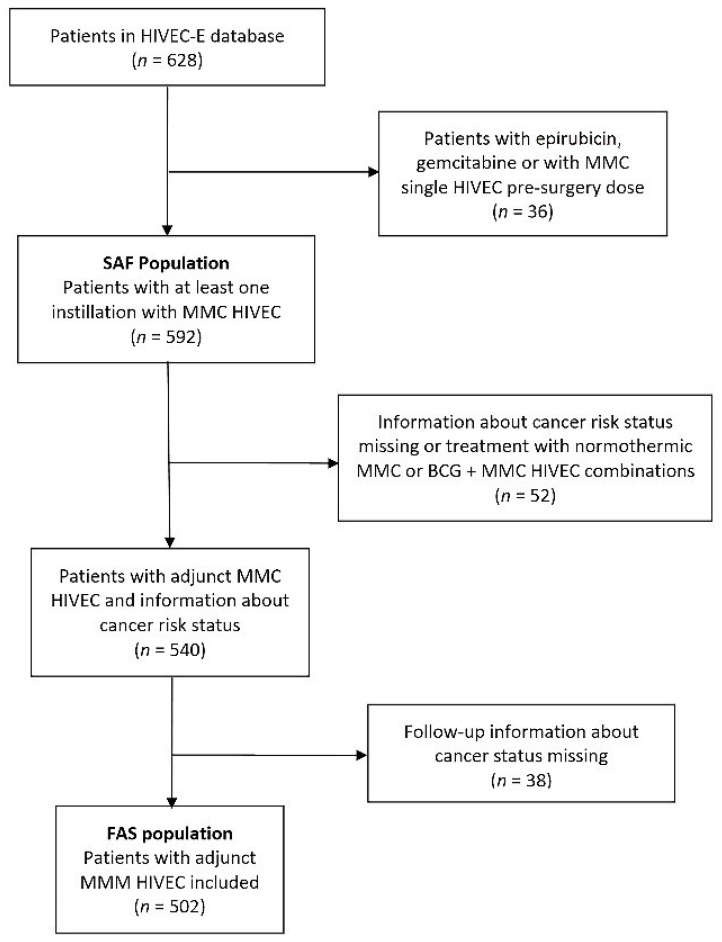
Flowchart of patients included in the study.

**Figure 2 jcm-10-05105-f002:**
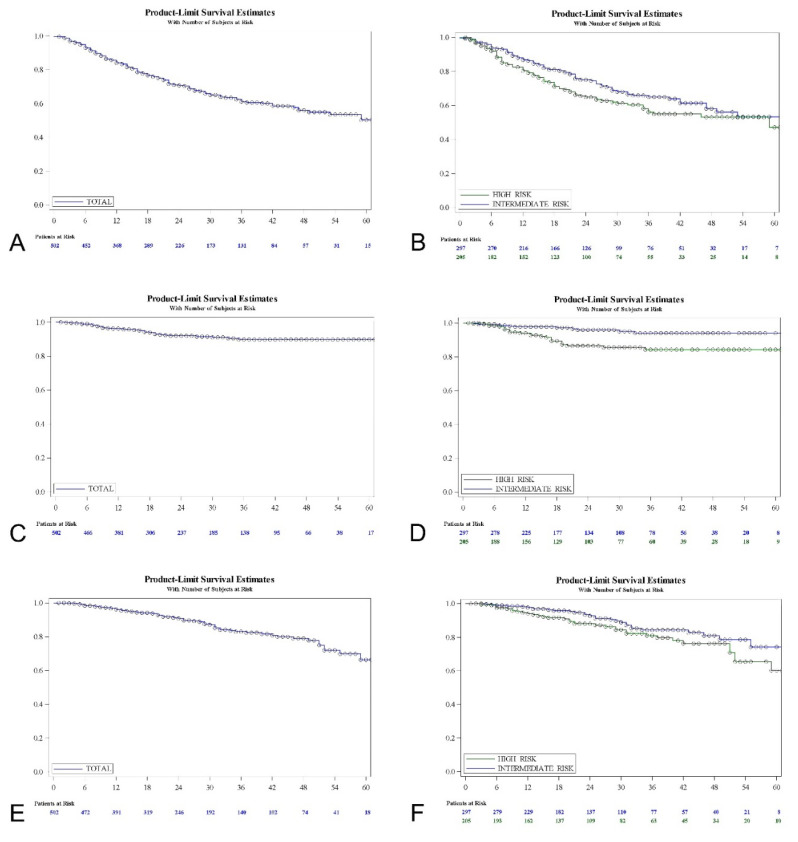
Recurrence-free survival, FAS population (**A**) and EAU risk groups (**B**); progression-free survival, FAS population (**C**) and EAU risk groups (**D**); overall survival, FAS population (**E**) and EAU risk groups (**F**).

**Figure 3 jcm-10-05105-f003:**
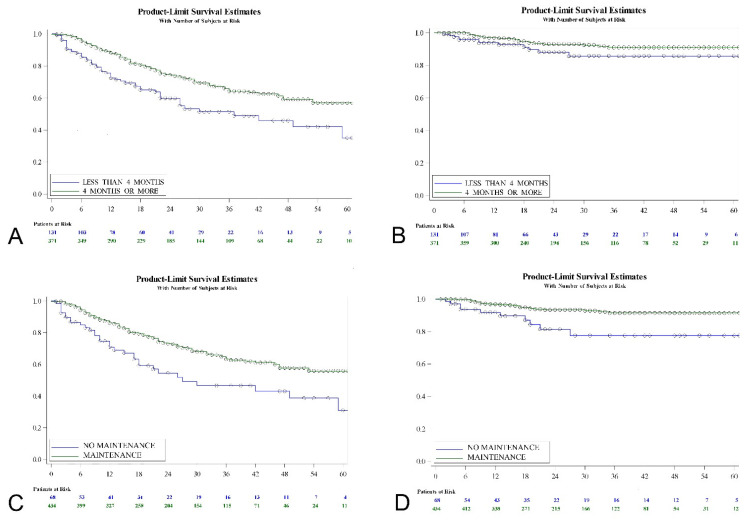
Recurrence (**A**) and progression-free survival (**B**) according to duration of adjunct HIVEC MMC treatment; recurrence (**C**) and progression-free survival (**D**) according to whether maintenance schedule was used.

**Table 1 jcm-10-05105-t001:** Clinico-pathological characteristics of patients, FAS population (*n* = 502).

Variable	*n* (%)
Sex, *n* (%)	
Male	414 (82.5)
Female	88 (17.5)
Age, years *	69.6 ± 10.6 (34–94)
BMI, kg/m^2^ *	3.4 ± 1.3 (1–6)
Smoking status, *n* (%)	
Non-smoker	92 (18.3)
Ex-smoker	256 (51)
Current smoker	128 (25.5)
Unknown	26 (5.2)
Number of tumors, *n* (%) ^(#)^	
Single	258 (52.4)
Multiple	234 (47.6)
Tumor size, *n* (%) ^(#)^	
<3 cm	333 (67.7)
≥3 cm	159 (32.3)
Pathological stage, *n* (%)	
Ta	376 (74.9)
T1	116 (23.1)
Primary carcinoma in situ	10 (2)
Grade ^(##)^, *n* (%)	
G1	173 (34.45)
G2	178 (35.45)
G3	151 (30.1)
EAU Risk stratification, *n* (%)	
Intermediate-risk	297 (59.2)
High-risk	205 (40.8)
Previous treatment with MMC, *n* (%)	69 (13.7)
Previous treatment with BCG, *n* (%)	51 (10.15)
Follow-up, months *	24.45 ± 16.5 (1–81)
Recurrence during follow-up, *n* (%)	159 (31.7)
Progression during follow-up, *n* (%)	35 (7)
Overall mortality, during follow-up (%)	66 (13.15)

* Values expressed in mean ± SD (range); BMI, body mass index; ^(#)^ excluding carcinoma in situ; ^(##)^, Grade according to WHO; MMC, mitomycin; BCG, bacillus Calmette-Guérin.

**Table 2 jcm-10-05105-t002:** Recurrence, progression and overall mortality at different times with interval limits for the FAS population (*n* = 502), and for intermediate (*n* = 297) and high-risk patients (*n* = 205).

Recurrence-Free Survival	Percent	95% CI	Log-Rank Test
Total series			
1 year	84.12	80.46–87.15	
2 years	70.72	66.03–74.89	
5 years	50.37	41.38–58–69	
Intermediate-risk			*p* = 0.075
1 year	86.77	82.11–90.28	
2 years	75.13	69.00–80.22	
5 years	53.30	42.75–62.76	
High-risk			
1 year	80.34	73.99–85.29	
2 years	64.88	57.37–71.40	
5 years	47.14	33.44–59.67	
**Progression-free survival**			
Total series			
1 year	96.24	94.01–97.65	
2 years	91.97	88.69–94.31	
5 years	89.83	85.81–92.75	
Intermediate-risk			*p* = 0.001
1 year	97.79	95.14–99.00	
2 years	95.99	92.27–97.94	
5 years	94.02	88.87–96.83	
High-risk			
1 year	93.99	89.41–96.63	
2 years	86.52	80.16–90.95	
5 years	84.23	77.02–89.34	
**Overall survival**			
Total series			
1 year	96.23	94–97.64	
2 years	90.8	87.34–93.35	
5 years	66.35	54.67–75.68	
Intermediate-risk			*p* = 0.064
1 year	97.73	95–98.97	
2 years	92.73	88.07–95.62	
5 years	74.26	60.55–83.82	
High-risk			
1 year	94.09	89.56–96.68	
2 years	88.09	82.06–92.19	
5 years	60.12	43.45–73.29	

**Table 3 jcm-10-05105-t003:** Cox regression model to predict tumor recurrence, FAS population (*n* = 502).

**Univariate Analysis**	**Hazard Ratio**	**95% CI**	***p*-Value**
High- vs. intermediate-risk group	1.322	0.968–1.805	0.0783
T1 vs. Ta	1.44	1.019–2.037	0.0009
Primary cis vs. T1	2.609	1.177–5.784	
Primary cis vs. Ta	3.758	1.746–8.088	
G2 vs. G1	1.088	0.733–1.615	0.1065
G3 vs. G2	1.347	0.927–1.956	
G3 vs. G1	1.466	1.003–2.145	
Cis vs. no cis	2.551	1.472–4.424	0.0008
Recurrent vs. primary	1.756	1.285–2.399	0.0004
Treatment duration < 4 vs. ≥4 months	1.842	1.321–2.568	0.0003
No maintenance vs. maintenance	1.929	1.307–2.846	0.0009
Previous MMC vs. no	1.611	1.071–2.424	0.0299
Previous BCG vs. no	1.834	1.187–2.84	0.0063
Age ≥ 70 vs. <70 years	1.254	0.917–1.715	0.156
Male vs. female	1.326	0.852–2.061	0.2102
Smoker vs. non-smoker	1.274	0.81–2.004	0.2948
Multiple vs. single tumor	1.220	0.888–1.675	0.2199
Size ≥ 3 vs. <3 cm	1.024	0.724–1.449	0.8895
**Multivariate analysis**	**Hazard Ratio**	**95% CI**	***p*-value**
Recurrent vs. primary	1.828	1.327–2.518	0.0002
Treatment duration < 4 vs. ≥4 months	1.724	1.235–2.407	0.0014
High- vs. intermediate-risk group	1.472	1.071–2.024	0.0171

**Table 4 jcm-10-05105-t004:** Cox regression model to predict tumor progression, FAS population (*n* = 502).

**Univariate Analysis**	**Hazard Ratio**	**95% CI**	***p*-Value**
High- vs. intermediate-risk group	3.076	1.506–6.289	0.002
T1 vs. Ta	3.159	1.596–6.253	0.0008
Primary cis vs. T1	2.078	0.477–9.045	
Primary cis vs. Ta	6.563	1.515–28.438	
G2 vs. G1	3.03	0.977–9.433	0.0059
G3 vs. G2	1.808	0.877–3.731	
G3 vs. G1	5.494	1.865–16.129	
Cis vs. no cis	4.424	1.718–11.363	0.0021
Recurrent vs. primary	2.876	1.431–5.784	0.003
Treatment duration < 4 vs. ≥4 months	1.908	0.948–3.84	0.0702
No maintenance vs. maintenance	3.07	1.474–6.396	0.0027
Previous MMC vs. no	2.561	1.199–5.468	0.0151
Previous BCG vs. no	2.717	1.233–5.988	0.0132
Age ≥ 70 vs. <70 years	1.681	0.847–3.339	0.1377
Male vs. female	1.062	0.441–2.557	0.893
Smoker vs. non-smoker	1.011	0.417–2.45	0.9797
Multiple vs. single tumor	1.158	0.591–2.269	0.6684
Size ≥ 3 vs. <3 cm	1.165	0.564–2.403	0.6788
**Multivariate Analysis**	**Hazard Ratio**	**95% CI**	***p*-value**
High- vs. intermediate-risk group	3.891	1.886–8	0.0002
Recurrent vs. primary	3.32	1.613–6.833	0.0011
No maintenance vs. maintenance	2.374	1.125–5.01	0.0233

**Table 5 jcm-10-05105-t005:** Cox regression model to predict tumor overall survival, FAS population (*n* = 502).

**Univariate Analysis**	**Hazard Ratio**	**95% CI**	***p*-Value**
High- vs. intermediate-risk group	1.572	0.967–2.557	0.0679
T1 vs. Ta	1.64	0.99–2.715	0.1579
Primary cis vs. T1	0	0	
Primary cis vs. Ta	0	0	
G2 vs. G1	1.801	0.935–3.46	0.0533
G3 vs. G2	1.228	0.712–2.118	
G3 vs. G1	2.212	1.158–4.219	
Cis vs. no cis	1.2	0.375–3.831	0.7571
Recurrent vs. primary	1.017	0.618–1.675	0.947
Treatment duration <4 vs. ≥4 months	1.948	1.171–3.239	0.0102
No maintenance vs. maintenance	1.789	0.984–3.253	0.0565
Previous MMC vs. no	0.967	0.461–2.024	0.9294
Previous BCG vs. no	0.801	0.29–2.217	0.6709
Age ≥70 vs. <70 years	3.459	1.943–6.157	<0.0001
Male vs. female	1.663	0.793–3.496	0.1781
Smoker vs. non-smoker	0.973	0.506–1.869	0.9353
Multiple vs. single tumor	1.158	0.591–2.269	0.6684
Size ≥3 vs. <3 cm	1.024	0.724–1.449	0.8895
**Multivariate Analysis**	**Hazard Ratio**	**95% CI**	***p*-value**
Age ≥70 vs. <70 years	3.356	1.884–5.976	<0.0001
Treatment duration <4 vs. ≥4 months	1.824	1.095–3.039	0.0211

**Table 6 jcm-10-05105-t006:** Frequency and severity of adverse events (AEs) for FAS population (*n* = 592) receiving at least one instillation of HIVEC MMC.

	Grade 1–2 *n* (%)	Grade 3–4 *n* (%)	Total *n* (%)
Dysuria	59 (34.7)	0 (0)	59 (9.9)
Frequency	5 (2.9)	0 (0)	5 (0.8)
Irritative symptoms	18 (9.4)	1 (6.25)	17 (2.9)
Bladder Pain	37 (21.8)	5 (31.25)	42 (7.1)
Urgency	34 (20)	0 (0)	34 (5.7)
Urinary retention	0 (0)	0 (0)	0 (0)
Bladder spasms	21 (12.35)	1 (6.25)	22 (3.7)
Bacterial cystitis	11 (6.5)	1 (6.25)	12 (2)
Renal colic	1 (0.6)	0 (0)	1 (0.2)
Hematuria	18 (10.6)	3 (18.75)	21 (3.55)
Incontinence	8 (4.7)	1 (6.25)	9 (1.5)
Fever	2 (1.2)	1 (6.25)	3 (0.5)
Flu-like symptoms	0 (0)	0 (0)	0 (0)
Skin rash	27 (15.9)	2 (12.5)	29 (4.9)
Other AEs	7 (4.1)	3 (18.75)	12 (2)
Total number EAs	248 (100)	18 (100)	266 (100)
Number of patients with AES	170	16	186
Number of patients w/o AEs	-	-	406
SAF population	-	-	592

AEs, Adverse effects; SAF, safety population.

## Data Availability

Full data will be provided upon a reasonable request to the corresponding author.

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
