# Peer review of "Long-Term Experience with Hyperthermic Chemotherapy (HIVEC) Using Mitomycin-C in Patients with Non-Muscle Invasive Bladder Cancer in Spain"

_jcm, 2021, doi:10.3390/jcm10215105_

Round 1

Reviewer 1 Report

It is a multicenter study on highly interesting topic of adjuvant treatment in NMIBC patients. The problem of BCG shortage appears to be even greater in the Covid-19 pandemic, so the challenges in the optimalization of intravesical chemotherapy instillations is of supreme importance.

Some minor points for considerations / comments to be included were listed below:

  • Was the trial registered at respective website? Can you provide the details in the Material and methods?
  • There were approximately 500 patients included, which gives about <10pts/center/year (2012-2020, 9 centers). Taking into consideration the predominance of intermediate and highrisk cases of NMIBC, this low number should be commented. Were the rest of the patients managed in the clinics treated with no adjuvant or BCG (10% received BCG before inclusion)?
  • There were mainly small (< 3 cm) and Ta included (table 1), while grading indicated intermediate/high risk tumors based on EAU stratification (Table 1). This could lead to different conclusions when focused on greater or multiple tumors.
  • The maintenance of treatment was short – as 70% of patients reached 4 months.
  • Please increase the quality/resolution of figures.
  • The sensitivity of CIS was excellent yet on a small group of patients (n=10). As the instillation are the only solution for the bladder saving option, it should be further studied. Please kindly discuss the influence of chemotherapy on cis, and the role in the reduction of progression /recurrence. This is perceived as a major difference when BCG is compared to standard intravesical chemotherapy.
  • Please check for typing errors (e.g. table 5 line 285, overll; no ) at the end of line 407).
  • Please cite lines 316 and 382 in discussion.

Author Response

Thank you very much, please see word file included (Response to Reviewer #1)

Reviewer 2 Report

  1. Introduction is very long. The second paragraph about shorten of bcg is redundant no need to repeat.
  2. S. there also a global shortage in MMC due to insufficient sterility….
  3. There is no evidence base study that verified what is the best maintenance schedule to MMC, how did you choose your schedule.
  4. I could not find how many of the recruited patients failed previous BCG treatment, this is group of patients has maximum beneficial from thero-chemotherapy.

Author Response

Thank you very much. Please see word file enclosed (Response to Reviewer #2).
